# Consumption of Food Supplements during the Three COVID-19 Waves in Poland—Focus on Zinc and Vitamin D

**DOI:** 10.3390/nu13103361

**Published:** 2021-09-25

**Authors:** Anna Puścion-Jakubik, Joanna Bielecka, Monika Grabia, Anita Mielech, Renata Markiewicz-Żukowska, Konrad Mielcarek, Justyna Moskwa, Sylwia K. Naliwajko, Jolanta Soroczyńska, Krystyna J. Gromkowska-Kępka, Patryk Nowakowski, Katarzyna Socha

**Affiliations:** Department of Bromatology, Faculty of Pharmacy with the Division of Laboratory Medicine, Medical University of Białystok, Mickiewicza 2D Street, 15-222 Białystok, Poland; joanna.bielecka@umb.edu.pl (J.B.); monika.grabia@umb.edu.pl (M.G.); anita.mielech@umb.edu.pl (A.M.); renmar@poczta.onet.pl (R.M.-Ż.); konrad.mielcarek@umb.edu.pl (K.M.); justyna.moskwa@umb.edu.pl (J.M.); sylwia.naliwajko@umb.edu.pl (S.K.N.); jolanta.soroczynska@umb.edu.pl (J.S.); krystyna.gromkowska.kepka@gmail.com (K.J.G.-K.); patryk.nowakowski@umb.edu.pl (P.N.); katarzyna.socha@umb.edu.pl (K.S.)

**Keywords:** food supplements, immunity, COVID-19, zinc, vitamin D, lifestyle, Poland

## Abstract

Food supplements (FS) are a concentrated source of vitamins, minerals, or other ingredients with nutritional or other physiological effects. Due to their easy availability, widespread advertising, and sometimes low price, increased consumption of this group of preparations has been observed. Therefore, the aim of the study was to assess the knowledge and intake of FS during the COVID-19 pandemic in Poland, with particular reference to FS containing zinc and vitamin D. It was noted that both of the above ingredients were used significantly more often by people with higher education (59.0%), with a medical background or related working in the medical field (54.5%), and/or exercising at home (60.1%). Preparations containing vitamin D were used by 22.8% of the respondents in the first wave, 37.6% in the second wave, and 32.9% in the third wave. To sum up, we showed the highest consumption of vitamin and mineral supplements, and preparations containing zinc and vitamin D were taken significantly more often by people with higher medical and related education. This indicates a high awareness of health aspects and the need for preventive measures in these groups.

## 1. Introduction

According to the definition of the European Food Safety Authority (EFSA), a food supplement (FS) is a foodstuff intended to be a complement to a normal diet and is a concentrated source of nutrients (vitamins, minerals) or other substances with a nutritional or physiological effect. FS could contain specific substances separately or in a complex combination. There are different forms of FS: pills, tablets, capsules, powder sachets, liquid ampoules, dropper bottles, and other. In the European Union, FS are regulated as foods and must be safe to consume. The regulations determine the maximum level for vitamins and minerals and some other substances. However, in Poland only maximum levels of vitamins and minerals are determined. FS could be used to correct nutritional deficiencies or to maintain an adequate intake of nutrients. Their intake is not a substitute for a varied and balanced diet [1]. Prevention and treatment are physiological activities that are not allowed to be attributed to FS. Moreover, the label must not state that they are recommended for the treatment of diseases. 

In Poland, the first case of SARS-CoV-2 infection was diagnosed on 4 March 2020. One week later, the World Health Organization declared the COVID-19 pandemic (on 11 March 2020). To date, more than 209 million COVID-19 cases and over 4.4 million deaths have been reported worldwide, while in Poland over the total is 2.8 million cases and 75,000 deaths (as of 23 August 2021). Three waves of the pandemic can be distinguished in Poland so far. The first was between March and May 2020, the second between September and November 2020, and the third from February to April 2021. Most of the infected patients experienced mild to moderate symptoms such as tiredness, fever, cough, headache, and loss of smell or taste. However, the list of possible symptoms is getting longer due to the research being conducted on this topic. There is a higher risk of severe COVID-19 among the elderly and those suffering from chronic illnesses (e.g., diabetes, cancer, and chronic respiratory diseases) [2]. 

Adequate diet and nutritional status are key elements in the maintenance of the proper functioning of the immune system. SARS-CoV-2 infection is usually associated with a decreased immune response, leading to pneumonic inflammation. Among the notably important components improving immune functions, vitamins (A, B_6_, B_9_, B_12_, C, D, and E), microelements (Fe, Cu, Se, and Zn), and n-3 long-chain polyunsaturated fatty acids (PUFAs) are indicated [3]. It was demonstrated that vitamin D supplementation was related to a lower risk of respiratory infections [4]. In a randomized trial, it was observed that humoral immunity was improved by the supplementation of vitamins A and D among pediatric patients who received influenza vaccination [5]. However, there is still insufficient evidence to provide exact recommendations on vitamin C, D, and E supplementation for the prevention and treatment of COVID-19 [6]. The positive influence of vitamin B_6_ on immunity involves activation of innate or adaptive immunity and the influence on the proliferation of immune cells [7]. Zn is important for the development and functioning of neutrophils and natural killer cells [8]. Fe modulates the differentiation and proliferation of T-cells and the production of reactive oxygen species, which take part in removing infectious agents. The influence of PUFAs on viral infections is not well established and requires further research [9]. Sufficient Se intake supports the immune system, while Se deficiency impairs innate and acquired immunity by the negative influence on cellular as well as humoral immunity (i.e., the production of antibodies) [10]. 

During the pandemic, negative changes in eating habits and lifestyle such as increased consumption of alcohol, sweets, and fast food or reduced physical activity were reported [11,12]. Taking into account the positive aspects, an increased intake of fruits, vegetables, nuts, legumes, and fish was also observed [13]. On the other hand, greater interest in searching for information on improving the immune system by food products or FS was observed [14,15].

In Poland, before the pandemic (in 2017), the worth of the FS market was estimated at 4.4 billion PLN (approximately 113 million USD) and over 70% of Poles used FS. It is estimated that in 2025 the global supplements market will reach 300 billion USD [16].

Currently, several anti-COVID-19 vaccines have been approved for use in humans to protect against the disease. However, vaccination hesitancy or resistance is observed among different populations [17]. Therefore, supporting immunity through adequate nutrition rich in essential nutrients and developing effective therapy against COVID-19 still seem to be crucial. 

This study aimed at an assessment of the changes in the FS intake patterns, with a special focus on the supplements influencing immunity during the three waves of the COVID-19 pandemic in Poland. 

## 2. Materials and Methods

### 2.1. Participants

This study was carried out among 935 Polish residents during three pandemic waves: *n* = 236 people answered questions about the first waves of the pandemic; the second: *n* = 364, and the third: *n* = 335. Each survey was conducted for about one month after the end of the period it covered. The study was conducted from July 2020 to April 2021. Responses of people living abroad (*n* = 9) were rejected. The inclusion criteria were being a resident of Poland, an adult (over 18 years of age), and answering all the questions. Each participant was informed that their participation was completely voluntary, the questionnaire was anonymous, and they could resign from participation in the study at any time. The researchers did not collect any data that could be used to identify people, including personal data. Each participant was allowed to complete the questionnaire only once. Consent to participate in the study was expressed by writing down the responses and sending them to the researchers.

### 2.2. Questionnaire

The questionnaire (containing questions and answers) was included as an attachment to the publication. The three questionnaires contained the same questions, but the third contained one additional question concerning the respondents’ knowledge about the possibility of preventing viral infections (number 35) (Appendix A).

### 2.3. Statistical Analysis

Statistical analysis of the results was performed using Statistica software (TIBCO Software Inc., Palo Alto, CA, USA) and calculator for the chi-square test [18]. The dependencies between the qualitative features were assessed using the Chi-square test of independence. The level of significance was *p* < 0.05.

## 3. Results

Most of the respondents were women (during the first wave: 80.0%, during the second wave: 81.9%, and during the third wave: 79.7%). Our survey was anonymous and voluntary, and we had no option to select a gender group. A larger percentage of women participating in the study may indicate, at the same time, greater interest in aspects of health and social life among people of this gender.

Residents of all 16 voivodeships participated in the study, but the vast majority were inhabitants of Podlaskie and Mazowieckie voivodeships; the remaining inhabitants accounted for less than 5%.

Adults with an average age of 31 ± 11, 28 ± 9, and 28 ± 10 years, mainly with higher education (66.9%, 59.3%, and 47.5%, respectively), participated in the survey. Most respondents lived in a large city of over 250,000 inhabitants (37.3%, 40.1%, and 36.1%) or a village (33.9%, 24.9%, and 30.7%). About half of the respondents from each group described their financial situation as rather good (56.8%, 53.3%, and 49.5%), and most households were comprised of 2–4 people. It is noteworthy that, during the first wave of the pandemic, as many as 50.0% of the respondents worked at their usual office or worksite, while during the second and third waves the percentage was lower (38.5% and 36.4%). About three-quarters of the respondents described their level of physical activity as low during each of the three periods (68.2%, 78.8%, and 78.2%) (Table 1).

It was shown that, during the first round of the survey, the highest percentage of respondents described their health as very good (39.8% vs. 27.0% and 21.5%). In the first half of 2020, the respondents significantly more often answered that they did not suffer from COVID-19. It is disturbing that, during the third wave, as many as 40.0% of respondents noticed an increase in their body weight, and only 42.7% undertook physical activity at home (Table 2).

It is satisfactory that almost all respondents correctly answered what a dietary supplement is—that it only supplements nutritional deficiencies (99.2%, 100.0%, and 97.6%)—and know the difference between FS and medications; this answer was indicated by 91.5%, 97.5%, and 95.8%. It is surprising that the most frequently chosen category of FS during all three waves of the pandemic in Poland was vitamin and mineral preparations (40.3%, 60.2%, and 54.3%), and preparations affecting immunity came in second place. It was shown that preparations from this category were consumed by twice as many people during the second and third wave than during the first wave (18.2%, 37.4%, and 34.9%) (Table 3).

During the first wave, a significantly greater percentage of respondents declared not taking food supplements with zinc and vitamin D (63.6% vs. 30.0% and 39.4%), and the most important reason cited for using them was the desire to supplement deficiencies of vitamins and minerals (36.0%, 27.5%, and 54.3%) (Table 3).

The authority of pharmacists’ recommendations was noticeable during the second wave—as many as 13.5% of respondents chose these preparations at the recommendation of a pharmacist. An important fact is that the vast majority (over 90%) drink FS and medications with water. As many as 62.3% of respondents declared that they had not noticed an increase in the number of advertisements for FS during the pandemic. The vast majority of respondents used supplements as recommended (58.1%, 70.5%, and 63.6%). More than 85% of respondents (86.9%, 94.2%, and 88.4%) were aware of the side effects, and over 90% of the risk of overdose (91.1%, 96.1%, and 94.0%). It should also be emphasized that over 70% of respondents indicated that FS should be used only in the case of diagnosed deficiencies (70.8%, 77.5%, and 73.4%). A significantly higher percentage (95.9%) indicated an awareness of interactions between FS and medications in the second wave. In the third round of the survey, a question was added regarding awareness of the beneficial effect of the use of preparations containing zinc and vitamin D in the prevention of viral infections—79.7% of respondents indicated that they had heard such reports (Table 3).

In the following part, the entire study group was divided in terms of the use of FS containing only zinc, only vitamin D, or both. FS with both ingredients were chosen significantly more often by people with higher education (59.0%) and with medical and related education (54.5%) (Table 3). Among the inhabitants of large cities—with a population of over 250,000—the highest percentage of respondents used preparations containing both vitamin D and zinc. Preparations containing only zinc were significantly more often used by people assessing their financial situation as rather good (64.7%) and by students (70.6%). Four-person families used both these components (73.9%) as prophylaxis. The highest percentage of people who suffered from COVID-19 consumed both zinc and vitamin D (no statistical significance); however, the degree of dependence, i.e., whether these ingredients were used before or after infection, was not found (Table 4).

High health awareness may be indicated by the fact that people choosing both ingredients in food supplements were also active and exercised at home (60.1%). People who took both zinc and vitamin D also used other vitamin and mineral ingredients (85.5% of the respondents) and other FS affecting immunity (55.5%). It is surprising that a large percentage of the respondents also took preparations supporting the appearance of hair, skin, and nails (41.8%). It should be emphasized that 100% of respondents using zinc took the preparations in accordance with the recommendations, and 94.1% indicated that food supplements should be used only in the case of proven deficiencies (Table 4).

In our research, we found that the highest percentage of people during all three waves used food supplements containing only vitamin D, while searches on Google Trends indicate that, during the first wave, information about zinc was more popular—the importance of vitamin D increased during the second wave (Figure 1).

## 4. Discussion

There are several dietary and lifestyle factors that could influence immunity in positive as well as negative ways. The COVID-19 pandemic has prompted people to search for natural methods to boost immunity, including FS usage. Several vaccines are currently approved for human use, but there is still a need to develop effective therapies to treat COVID-19 and alleviate the negative health consequences of the disease. 

Currently, there are quite a few studies available on the impact of the pandemic on lifestyles and nutrition. However, there are not many studies dealing with COVID-19 and dietary supplement intake. To the best of our knowledge, our study is the first to assess the consumption of dietary supplements in Poland during the three COVID-19 waves.

A Google Trends analysis showed that, in Poland, in relation to the coronavirus, the following terms were searched for: vitamin C, vitamin D, and *Glycyrrhiza glabra*; globally, there were also search terms such as vitamin K, selenium, zinc, garlic, onion, elderberry, lactoferrin, echinacea, and *Nigella sativa* L. Polish residents were trying to find antiviral properties for turmeric, garlic, and iodine as well as immune-boosting properties for fish oil [14]. 

Kamarli Altun et al. conducted a cross-sectional study among Turkish dietitians concerning the supplements, functional foods, and herbal medicines they used to protect themselves against SARS-CoV-2 infection. Nearly 90% of the study participants found that proper nutrition could affect the clinical course of the disease and almost all respondents (94.5%) declared FS intake. Less than half of dietitians (46.1%) started using herbal medicine, while nearly one-third included functional foods into the diet (34.9%). Fish oil was the most commonly chosen FS (81.9%). Women were twice as likely to use FS as men [19]. In this study we reported a lower prevalence of FS intake: most of the participants declared usage of FS 57.2%, 80.3%, and 76.1% in the first, second, and third waves, respectively, of the pandemic in Poland. The most often chosen type of FS was preparations with vitamins and minerals. More people reported that pandemic affect on using these supplements in the second wave compared to the first and third waves (23.1% vs. 8.9% and 17.0%).

Another cross-sectional study was carried out by Alfawaz et al. among Saudi Arabian residents, focused on changes in FS usage before the pandemic and during lockdown. Males tended to use FS (multivitamin, selenium, zinc, and vitamin D) more frequently than females. Among the subgroup of COVID-19 patients, men used more multivitamin and zinc supplements than women, while women had a higher intake of supplements with vitamins D and C. The male study participants 26–35 years of age declared a significantly higher use of multivitamin supplements than females (30.1 vs. 22.6%; *p* < 0.054) of the same age group. As determinants of FS usage, researchers distinguished the influence of age, level of education, and income [20]. 

The supplementation pattern among COVID-19 patients in Teheran was analyzed by Bagheri et al. Significantly higher vitamin D intake was reported in outpatients (30%) compared to hospitalized patients (16.5%). It was observed that vitamin D intake was related to a reduced risk of exacerbation of the disease. Moreover, a relevant difference was found considering zinc intake—9% vs. 2% in outpatients and inpatients, respectively. However, none of the patients declared the usage of multivitamins, vitamin C, vitamin E, folic acid, iron, omega-3, and omega-6 fatty acids [21]. In this research, none of the participants had COVID-19 in the first wave, 11.8% in the second, and 17.6% in the third waves; 14%, 28.3%, and 33.7% of respondents could not equivocally say whether they had SARS-CoV-2 infection. Considering the supplementation of vitamin D and zinc at the beginning of the pandemic, most of the participants (63.6%) did not take them during the first wave, which is contrary to the responses about the second (30.1%) and third (39.4%) waves of the pandemic. This difference was statistically significant. Vitamin D intake was declared by 22.8%, 37.6%, and 32.9%, while zinc was taken by 1.3%, 1.9%, and 2.1%; both compounds were used by 7.2%, 14.0%, and 12.5% of participants during the three waves of COVID-19 in Poland. Our results indicate a higher prevalence of intake of FS with vitamin D and zinc among the Polish population than among the Iranian population.

While in many countries increased interest in diet supplementation was observed, the findings of the cross-sectional study among the Lebanese population showed a decreased supplement intake. Before the pandemic, over 73% of the respondents used FS, while after the COVID-19 outbreak it was 69.9%. However, for specific subgroups of FS, increased intake was reported. Noticeably higher usage of antioxidants (14% vs. 15.6%), vitamin C (35.3% vs. 42.1%), vitamin D (35.5% vs. 41%), vitamin E (15.2% vs. 17.5%), and zinc (18.8% vs. 29.3%) was reported [22]. Our results indicate that the COVID-19 pandemic breakout did not generally influence the pattern of supplementation among Polish residents. The vast majority of the respondents (90.7%, 61.8%, and 81.8% in the three waves, respectively) did not change their FS usage. On the other hand, we found that if the study participants decided to modify something in their diet supplementation, they tended to use more preparations, especially during the second wave (23.1%). 

Another analysis considering supplementation patterns during the pandemic was carried out on the basis of the results of the application-based community survey. This study involved 175,652 supplement users and 197,068 nonusers. The risk of COVID-19 infection among women who declared intake of probiotics, omega-3 fatty acids, multivitamins, and vitamin D was lowered by 14%, 12%, 13%, and 9%, respectively. No protective association was observed among men. Moreover, no positive effect was found for respondents taking vitamin C, zinc, or garlic FS [23]. 

Vitamin D supplementation had a positive influence on recovery from symptoms in patients with mild to moderate COVID-19. Sabico et al., in a randomized control trial, administrated two weeks of oral supplementation of vitamin D (1000 UI vs. 5000 UI) to patients with suboptimal vitamin D status. In the group that received a higher dose, a reduced time to recovery from cough and sensory loss was found. Based on these findings, it seems reasonable to recommend vitamin D as an adjuvant to COVID-19 therapy for patients with mild to moderate symptoms [24].

Vitamin D supplementation and the risk of COVID-19 were assessed in a prospective study by Hao et al. based on data from the UK Biobank cohort study. Habitual use of vitamin D supplements was related to a 34% lower risk of infection [25]. 

Szarpak et al. carried out a meta-analysis of four studies, comprising 1474 patients, focusing on the influence of zinc on COVID-19 patient outcomes. In the group of patients who received zinc supplementation, survival to hospital discharge was 56.8%, while in the group to which supplementation was not administered it was 75.9%. Moreover, patients who were given supplementation had a higher percentage of in-hospital mortality (22.3% vs. 13.6%) and longer hospital stay (7.7 days vs. 7.2 days). Based on these findings, zinc supplementation does not have a beneficial impact on the abovementioned outcomes [26]. Dubourg et al. observed that median blood Zn levels were significantly lower in COVID-19 patients with poor clinical outcomes in comparison to patients with good clinical outcomes (840 µg/L vs. 970 µg/L). Those results may indicate the importance of Zn supplementation during SARS-CoV-2 infection [27]. 

A positive correlation was shown between Zn deficiency and COVID-19 cases per million among Asian countries in a retrospective study by Ali et al. The prevalence of Zn deficiency was nearly twice as high among Asians compared to the European population (17.5% vs. 8.9%). On the other hand, a significantly negative correlation between serum Zn levels and COVID-19 deaths per million was recognized among the European population [28]. Hoverer, cohort studies are needed to confirm these observations. 

In research conducted by Adbelmaksoud et al., Zn supplementation (220 mg of zinc sulfate twice a day) was related a shortened time of smell recovery after SARS-CoV-2 infection, without an influence on the total recovery of the disease. Moreover, serum Zn levels were similar considering subgroups in the case of disease severity or the presence or absence of olfactory and/or gustatory dysfunction [29]. 

Thomas et al. carried out a randomized control trial among ambulatory patients (*n* = 214) suffering from COVID-19. Patients were randomized in a 1:1:1:1 allocation ratio every 10 days. In the first intervention group, patients were given 50 mg of zinc gluconate, the second 8000 mg of ascorbic acid, the third both, and in the last group a standard treatment regimen was observed. Researchers did not observe a significant difference in secondary outcomes among the studied groups. The results of the study by Thomas et al. do not confirm the assumption of the study by Dubourg et al. [30].

It should be emphasized that the vast majority of respondents took dietary supplements, and not drugs containing zinc and vitamin D. Drugs were used by 5.1% of the respondents during the first wave; during the second wave it was 15.1%, and during the third 13.1%. The respondents themselves noticed the need for supplementation, and the advice of specialists (doctors, pharmacists) was much less frequently cited. Therefore, education on the proper selection of a preparation (the right chemical form, with good digestibility) and the right dose (in accordance with the recommendations corresponding to the daily requirement for supplemented dietary components) seems to be important. It was estimated that many respondents did not provide the names of the zinc and vitamin D preparations used—this may indicate that they do not pay attention to it, while one of the important criteria is the price of FS.

The limitations of this study include the unequal gender proportions (the predominance of women)—if the majority of participants in the study were men, the results could be different. However, greater female participation is a fairly common problem in volunteer-based surveys. Moreover, our survey was a retrospective study, so incorrect recall of information by survey participants may be an important problem.

## 5. Conclusions

The popularity of dietary supplements, especially vitamin and mineral supplements, is gradually increasing in Poland. During the COVID-19 pandemic, the consumption of dietary supplements containing zinc and vitamin D increased, especially among people with higher education, or medical and paramedical education, which indicates the increased awareness of this social group regarding pro-health prophylaxis. Due to the nonrestrictive registration procedures of dietary supplements, it seems necessary to educate consumers in terms of the selection of appropriate preparations, proper nutrition, and balanced supplementation.

## Figures and Tables

**Figure 1 nutrients-13-03361-f001:**
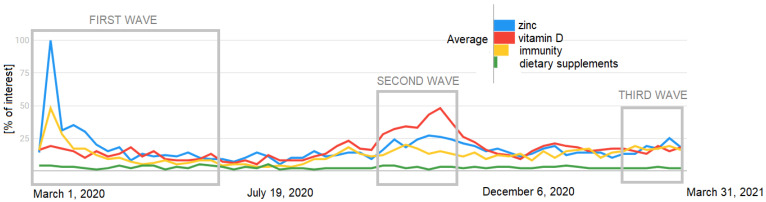
The popularity of searching for selected terms (own design based on data from Google Trends).

**Table 1 nutrients-13-03361-t001:** Characteristics of the study group.

Variable	First Wave *n* = 236 % (*n*)	Second Wave *n* = 364 % (*n*)	Third Wave *n* = 335 % (*n*)
**Gender**			
Female	80.0 (189)	81.9 (298)	79.7 (267)
Male	20.0 (47)	18.1 (66)	20.3 (68)
**Anthropometric measurements**			
Age (years)	31 ± 11	28 ± 9	28 ± 10
Mass (kg)	69 ± 15	69 ± 18	69 ± 15
Weight (m)	1.69 ± 0.08	1.70 ± 0.09	1.70 ± 10
BMI (kg/m^2^)	25.06 ± 4.41	23.91 ± 7.45	23.72 ± 4.45
**Education**			
Primary school	2.1 (5)	1.1 (4)	2.4 (8)
Higher	66.9 (158)	59.3 (216)	47.5 (159)
Secondary	31.0 (73)	39.6 (144)	50.1 (168)
**Type of education**			
Medical and related	47.5 (112)	56.6 (206)	45.1 (151)
Nonmedical	36.4 (86)	22.3 (81)	25.4 (85)
Not applicable	16.1 (38)	21.1 (77)	29.5 (99)
**Place of residence**			
City with up to 150,000 inhabitants	21.6 (51)	26.7 (97)	26.3 (88)
City with 150,000–250,000 inhabitants	7.2 (17)	8.3 (30)	6.9 (23)
City with over 250,000 inhabitants	37.3 (88)	40.1 (146)	36.1 (121)
Village	33.9 (80)	24.9 (91)	30.7 (103)
**Subjective assessment of the material situation**			
Very good	20.8 (49)	25.8 (94)	22.1 (74)
Average	20.8 (49)	20.1 (73)	26.0 (87)
Rather good	56.8 (134)	53.3 (194)	49.5 (166)
Rather bad	1.6 (4)	0.5 (2)	2.1 (7)
Bad	0.0 (0)	0.3 (1)	0.3 (1)
**Number of people in the household**			
1	4.2 (10)	7.5 (28)	7.8 (26)
2	24.6 (58)	22.8 (83)	20.6 (69)
3	29.2 (69)	25.3 (92)	19.7 (66)
4	24.6 (58)	30.5 (111)	29.9 (100)
5	10.6 (25)	10.2 (37)	13.7 (46)
6	2.5 (9)	2.2 (8)	2.7 (9)
7	2.5 (6)	0.3 (1)	3.6 (12)
8	0.5 (1)	0.8 (3)	1.2 (4)
10	0.0 (0)	0.3 (1)	0.9 (3)
**Professional activity**			
Unemployed person	7.2 (17)	1.9 (7)	3.6 (12)
Person working in office	50.0 (118)	38.5 (140)	36.4 (122)
Person working remotely	8.5 (20)	8.2 (30)	6.0 (20)
Student	34.3 (81)	51.4 (187)	54.0 (181)
**Physical activity**			
Inactivity (sedentary)	0.0 (0)	0.0 (0)	0.0 (0)
Low (occasional exercise, 1–3 times a week)	68.2 (161)	78.8 (287)	78.2 (262)
Moderate (1 h of exercise per day)	25.4 (60)	18.9 (68)	17.3 (58)
High (hard physical work and daily workouts)	6.4 (15)	2.3 (9)	4.5 (15)

**Table 2 nutrients-13-03361-t002:** Assessment of health and physical activity during COVID-19.

Variable	First Wave (*n* = 236)	Second Wave (*n* = 364)	Third Wave (*n* = 335)
**How would you rate your health at the beginning of the pandemic in Poland?**			
Very good	39.8 (94)	27.0 (98)	21.5 (72)
Good	45.3 (107)	58.2 (212)	61.8 (207)
Medium	11.4 (27)	12.9 (47)	14.6 (49)
Poor	3.5 (8)	1.9 (7)	2.1 (7)
**Have you had COVID-19?**			
Yes	0.0 (0)	11.8 (43)	17.6 (59)
No	86.0 (203) ***	59.9 (218)	48.7 (163)
It is difficult to say unequivocally	14.0 (33)	28.3 (103)	33.7 (113)
**Has your body weight changed during the pandemic?**			
No	48.3 (114)	49.5 (180)	43.9 (147)
Increased	37.3 (88)	31.9 (116)	40.0 (134)
Decreased	14.4 (34)	18.6 (68)	16.1 (54)
**Did you exercise at home during the pandemic?**			
Yes	42.4 (100)	48.4 (176)	42.7 (143)
No	57.6 (136)	51.6 (188)	57.3 (192)

Differences between the various pandemic waves: *** *p* < 0.001.

**Table 3 nutrients-13-03361-t003:** Assessment of knowledge about food supplements and their consumption during the COVID-19 pandemic.

Variable	First Wave (*n* = 236)	Second Wave (*n* = 364)	Third Wave (*n* = 335)
**What is a food supplement?**			
A preparation that treats nutritional deficiencies	0.8 (2)	0.0 (0)	2.4 (8)
A preparation that only replenishes nutritional deficiencies	99.2 (234)	100.0 (364)	97.6 (327)
**Do you think food supplements differ from medications?**			
Yes	91.5 (216)	97.5 (355)	95.8 (321)
No	2.9 (7)	2.2 (8)	2.4 (8)
I do not know	5.6 (13)	0.3 (1)	1.8 (6)
**What categories of food supplements did you use during the pandemic?#**			
Vitamin–mineral supplements	40.3 (95)	60.2 (219)	54.3 (182)
Probiotics	13.1 (31)	18.1 (66)	15.5 (52)
Prebiotics	3.4 (8)	2.7 (10)	3.3 (11)
Supporting immunity	18.2 (43)	37.4 (136)	34.9 (117)
Supporting weight loss	3.0 (7)	1.9 (7)	3.0 (10)
Improving the condition of the hair, skin, and nails	14.4 (34)	19.5 (71)	22.4 (75)
Supporting the functioning of the urinary tract	2.1 (5)	0.8 (3)	3.3 (11)
Supporting the heart	1.7 (4)	1.4 (5)	4.2 (14)
Supporting memory	2.5 (6)	4.7 (17)	8.4 (28)
Supporting lowering cholesterol levels	1.3 (3)	0.8 (3)	1.2 (4)
Vision support	1.7 (4)	3.0 (11)	3.3 (11)
Supporting the functioning of the joints	5.1 (12)	4.7 (17)	1.8 (6)
Relieving the symptoms of menopause	0.0 (0)	0.0 (0)	0.6 (2)
Supporting the digestive tract	3.8 (9)	3.8 (14)	5.4 (18)
Improving well-being	3.4 (8)	4.4 (16)	4.8 (16)
Facilitating sedation and sleep	6.8 (16)	6.9 (25)	11.3 (38)
Supporting libido	1.3 (3)	0.8 (3)	0.3 (1)
Supporting alcohol metabolism	0.4 (1)	1.4 (5)	0.3 (1)
For athletes	3.8 (9)	2.7 (10)	5.1 (17)
Removing excess water	0.4 (1)	0.0 (0)	0.0 (0)
Other	0.0 (0)	1.6 (6)	0.0 (0)
I did not use dietary supplements	42.8 (101)	19.7 (72)	23.9 (80)
**Have you used zinc and vitamin D food supplements since March 2020?**			
No	63.6 (150) **	30.0 (109)	39.4 (132)
Only drugs	5.1 (12)	16.5 (60)	13.1 (44)
Yes both	7.2 (17)	14.0 (51)	12.5 (42)
Only zinc	1.3 (3)	1.9 (7)	2.1 (7)
Only vitamin D	22.8 (54)	37.6 (137)	32.9 (110)
**Why did you use such food supplements?#**			
Not applicable	47.9 (113)	24.7 (90)	26.9 (90)
To improve health	22.2 (52)	21.4 (78)	33.7 (113)
Due to a pharmacist’s recommendation	1.7 (4)	13.5 (49)	3.0 (10)
Due to a doctor’s recommendation	3.8 (9)	0.0 (0)	0.3 (1)
To supplement deficiencies of vitamins and minerals	36.0 (85)	27.5 (100)	54.3 (182)
To supplement the therapy prescribed by doctor	2.5 (6)	9.9 (36)	10.1 (34)
Due to a friend’s recommendation	5.1 (12)	6.0 (22)	3.0 (10)
Because I was encouraged by TV/media/Internet advertising	0.0 (0)	0.3 (1)	1.2 (4)
Other	0.0 (0)	0.0 (0)	2.1 (7)
**What do you usually use to wash down food supplements and medications?#**			
Tea	8.1 (19)	11.3 (41)	22.4 (75)
Cola	1.7 (4)	0.5 (2)	1.5 (5)
Not applicable	0.8 (2)	4.7 (17)	6.0 (20)
I do not drink	5.1 (12)	0.5 (2)	1.2 (4)
Juice	3.8 (9)	4.7 (17)	4.8 (16)
Water	93.2 (220)	93.4 (340)	90.1 (302)
Coffee	0.4 (1)	2.5 (9)	4.5 (15)
Milk	0.4 (1)	0.0 (0)	0.9 (3)
Other	0.4 (1)	0.5 (2)	0.6 (2)
**Do you think there were more advertisements for food supplements during the pandemic?**			
No	5.1 (12)	1.4 (5)	3.6 (12)
Yes	32.6 (77)	42.9 (156)	59.1 (198)
I did not notice a change	62.3 (147) **	55.7 (203)	37.3 (125)
**Do you use food supplements in the amount recommended on the package?**			
I do not use it	39.9 (80)	19.0 (69)	23.0 (77)
No, I use lower doses	3.8 (9)	4.7 (17)	6.0 (20)
No, I use higher doses	4.2 (10)	5.8 (21)	7.5 (25)
Yes	58.1 (137)	70.5 (257)	63.6 (213)
**Do you think food supplements can have side effects?**			
No, taking them is absolutely safe	13.1 (31)	5.8 (21)	11.6 (39)
Yes	86.9 (205)	94.2 (343)	88.4 (296)
**How do you assess the advisability of using food supplements?**			
They should be used only in the event of identified deficiencies	70.8 (167)	77.5 (282)	73.4 (246)
Their use is unnecessary	13.1 (31)	7.7 (28)	9.0 (30)
I have no opinion	16.1 (38)	14.8 (54)	17.6 (59)
**Do you think food supplements can be overdosed on?**			
No, they’re safe	8.9 (21)	3.8 (14)	6.0 (20)
Yes	91.1 (215)	96.1 (350)	94.0 (315)
**Do you think that food supplements can interact with medications prescribed by your doctor, and thus affect the effectiveness of therapy?**			
No, they’re safe	10.2 (24)	4.1 (15)	9.0 (30)
Yes	89.8 (212)	95.9 (349) ***	91.0 (305)
**Has the pandemic affected your use of food supplements?**			
No	90.7 (214) ***	61.8 (225)	81.8 (274)
Yes, I use fewer	0.4 (1)	1.4 (5)	1.2 (4)
Yes, I use more	8.9 (21)	23.1 (84)	17.0 (57)
I did not use food supplements	0.0 (0)	13.7 (50)	0.0 (0)
**Have you heard that preparations containing zinc and vitamin D can support immunity and be helpful in the prevention of viral infections?#**			
No	-	-	20.3 (68)
Yes	-	-	79.7 (267)

Differences between the various pandemic waves: ** *p* < 0.01, *** *p* < 0.001, # multiple choice question.

**Table 4 nutrients-13-03361-t004:** Consumption of food supplements with zinc, vitamin D and both ingredients depending on various factors.

Variable	Only Zinc *n* = 17 % (n)	Only Vitamin D *n* = 301 % (n)	Both Food Supplements *n* = 110
**Gender**			
Female	88.2 (15)	84.4 (254)	77.3 (85)
Male	11.8 (2)	15.6 (47)	22.7 (25)
**Education**			
Primary school	11.8 (2)	1.0 (3)	0.9 (1)
Secondary	58.8 (10)	41.7 (126)	40.1 (45)
Higher	29.4 (5)	57.3 (172)	59.0 (64) ***
**Type of education**			
Medical and related	35.3 (6)	50.1 (152)	54.5 (60) *
Nonmedical	29.4 (5)	27.9 (84)	27.3 (30)
Not applicable	35.3 (6)	22.0 (65)	18.2 (20)
**Place of residence**			
A city with up to 150,000 inhabitants	23.5 (4)	29.6 (89)	20.9 (23)
A city with 150,000–250,000 inhabitants	17.6 (3)	8.3 (25)	3.6 (4)
A city with over 250,000 inhabitants	35.4 (6)	39.2 (118)	47.3 (52)
Village	23.5 (4)	22.9 (69)	28.2 (31)
**Subjective assessment of material situation**			
Very good	23.5 (4)	23.6 (71)	30.0 (33)
Rather good	64.7 (11) ***	51.8 (156)	18.3 (55)
Average	11.8 (2)	23.6 (71)	15.5 (17)
Rather bad	0.0 (0)	1.0 (3)	4.5 (5)
Bad	0.0 (0)	0.0 (0)	0.0 (0)
**Number of people in the household**			
1	5.9 (1)	6.6 (20)	1.3 (4)
2	23.5 (4)	26.6 (80)	10.6 (32)
3	11.8 (2)	22.6 (68)	9.3 (28)
4	41.1 (7)	28.6 (86)	73.9 (31)
5	11.8 (2)	10.2 (31)	4.3 (13)
6	0.0 (0)	1.7 (5)	0.3 (1)
7	0.0 (0)	1.7 (5)	0.3 (1)
8	5.9 (1)	1.0 (3)	0.0 (0)
9	0.0 (0)	0.7 (2)	0.0 (0)
10	0.0 (0)	0.3 (1)	0.0 (0)
**Professional activity**			
Unemployed person	0.0 (0)	3.7 (11)	0.9 (5)
Person working in office	29.4 (5)	39.9 (120)	42.7 (47)
Person working remotely	0.0 (0)	8.3 (25)	8.2 (9)
Student	70.6 (12) *	48.1 (145)	48.2 (49)
**Physical activity**			
Inactivity (sedentary)	0.0 (0)	0.0 (0)	0.0 (0)
Low (occasional exercise, 1–3 times a week)	70.5 (12)	81.4 (245)	72.7 (80)
Moderate (1 h of training per day)	23.6 (4)	14.6 (44)	20.9 (23)
High (hard physical work and daily workouts)	5.9 (1)	3.9 (12)	6.4 (7)
**How would you rate your health at the beginning of the pandemic in Poland?**			
Very good	17.6 (3)	27.6 (83)	28.2 (31)
Good	64.7 (11)	57.8 (174)	56.4 (62)
Medium	11.8 (2)	11.9 (36)	11.8 (13)
Poor	5.9 (1)	2.7 (8)	3.6 (4)
**Have you had COVID-19?**			
Yes	11.8 (2)	12.0 (36)	15.5 (17)
No	46.5 (8)	58.8 (177)	62.2 (69)
It is difficult to say unequivocally	41.7 (7)	29.2 (88)	21.8 (24)
**Has your body weight changed during the pandemic?**			
No	47.1 (8)	47.2 (142)	40.0 (44)
Increased	47.1 (8)	33.9 (102)	41.8 (46)
Decreased	5.8 (1)	18.9 (57)	18.2 (20)
**Did you exercise at home during the pandemic?**			
Yes	41.2 (7)	48.8 (147)	60.1 (67)*
No	58.8 (10)	51.2 (154)	39.9 (43)
**What is a food supplement?**			
A preparation that treats nutritional deficiencies	0.0 (0)	2.0 (6)	0.9 (1)
A preparation that supplements nutritional deficiencies	100 (17)	98.0 (295)	99.1 (109)
**Do you think food supplements differ from medications?**			
Yes	100 (17)	97.0 (292)	99.1 (109)
No	0.0 (0)	0.0 (0)	0.9 (1)
I do not know	0.0 (0)	3.0 (9)	0.0 (0)
**What categories of food supplements did you use during the pandemic?#**			
Vitamin–mineral supplements	88.2 (15)	79.4 (239)	85.5 (94)
Probiotics	11.8 (2)	22.6 (68)	28.2 (31)
Prebiotics	0.0 (0)	5.0 (15)	5.5 (6)
Supporting immunity	29.4 (5)	45.5 (137)	55.5 (61)
Supporting weight loss	0.0 (0)	2.7 (8)	2.7 (3)
Improving the condition of hair, skin, and nails	35.3 (6)	18.6 (56)	41.8 (46)
Supporting the functioning of the urinary tract	5.9 (1)	3.0 (9)	2.7 (3)
Supporting the heart	5.9 (1)	3.7 (11)	5.5 (6)
Supporting memory	5.9 (1)	6.3 (19)	9.1 (10)
Supporting lowering cholesterol levels	0.0 (0)	2.0 (6)	1.8 (2)
Vision support	5.9 (1)	2.0 (6)	10.0 (11)
Supporting the functioning of the joints	5.9 (1)	5.3 (16)	7.3 (8)
Relieving the symptoms of menopause	0.0 (0)	0.0 (0)	0.9 (1)
Supporting the digestive tract	5.9 (1)	4.3 (13)	6.4 (7)
Improving well-being	0.0 (0)	5.0 (15)	9.1 (10)
Facilitating sedation and sleep	5.9 (1)	11.3 (34)	12.7 (14)
Supporting libido	0.0 (0)	0.7 (2)	1.8 (2)
Supporting alcohol metabolism	0.0 (0)	1.7 (5)	0.9 (1)
For athletes	5.9 (1)	4.3 (13)	6.4 (7)
Removing excess water	0.0 (0)	0.0 (0)	0.0 (0)
Other	0.0 (0)	0.3 (1)	0.0 (0)
I did not use food supplements	0.0 (0)	0.0 (0)	0.0 (0)
**Why did you use such food supplements?#**			
Not applicable	0.0 (0)	0.0 (0)	0.0 (0)
To improve health	35.3 (6)	54.8 (165)	54.5 (60)
Due to a pharmacist’s recommendation	17.6 (3)	3.0 (9)	5.5 (6)
Due to a doctor’s recommendation	17.6 (3)	0.3 (1)	0.0 (0)
To supplement deficiencies of vitamins and minerals	52.9 (9)	72.1 (217)	69.1 (76)
To supplement the therapy prescribed by doctor	0.0 (0)	13.3 (40)	14.5 (16)
Due to a friend’s recommendation	5.9 (1)	6.6 (20)	0.9 (1)
Because I was encouraged by TV/media/Internet advertising	0.0 (0)	0.7 (2)	0.9 (1)
Other	0.0 (0)	1.0 (3)	1.8 (2)
**What do you usually use to wash down food supplements and medications?#**			
Tea	11.8 (2)	13.3 (40)	18.2 (20)
Cola	5.9 (1)	0.6 (2)	1.8 (2)
Not applicable	0.0 (0)	0.0 (0)	0.0 (0)
I do not drink	0.0 (0)	0.0 (0)	6.4 (7)
Juice	11.8 (2)	5.3 (16)	0.0 (0)
Water	94.1 (16)	98.0 (295) ***	97.3 (107)
Coffee	17.6 (3)	3.3 (10)	0.9 (1)
Milk	0.0 (0)	0.0 (0)	0.0 (0)
Other	0.0 (0)	0.0 (0)	0.0 (0)
**Do you think there were more advertisements for food supplements during the pandemic?**			
No	0.0 (0)	2.7 (8)	0.9 (1)
Yes	35.3 (6)	41.5 (125)	37.3 (41)
I did not notice a change	64.7 (11)	55.8 (168)	61.8 (68)
**Do you use food supplements in the amount recommended on the package?**			
I do not use it	0.0 (0)	0.0 (0)	0.0 (0)
No, I use lower doses	0.0 (0)	6.3 (19)	7.3 (8)
No, I use higher doses	0.0 (0)	8.3 (25)	13.6 (15)
Yes	100.0 (17) ***	85.4 (257)	79.1 (87)
**Do you think food supplements can have side effects?**			
No, taking them is absolutely safe	11.8 (2)	10.3 (31)	10.0 (11)
Yes	88.2 (15)	89.7 (270)	90.0 (99)
**How do you assess the advisability of using food supplements?**			
They should be used only in the event of identified deficiencies	94.1 (16) **	82.1 (247)	83.6 (92)
Their use is unnecessary	5.9 (1)	5.0 (15)	0.9 (1)
I have no opinion	0.0 (0)	12.9 (39)	15.5 (17)
**Do you think food supplements can be overdosed on?**			
No, they’re safe	11.8 (2)	5.3 (16)	7.3 (8)
Yes	88.2 (15)	94.7 (285)	92.7 (102)
**Do you think that food supplements can interact with medications prescribed by your doctor, and thus affect the effectiveness of therapy?**			
No, they’re safe	23.6 (4)	8.0 (24)	10.9 (12)
Yes	76.4 (13)	92.0 (277) **	89.1 (98)
**Has the pandemic affected your use of food supplements?**			
No	76.5 (13)	73.7 (222)	59.1 (65)
Yes, I use fewer	0.0 (0)	0.7 (2)	1.8 (2)
Yes, I use more	23.5 (4)	25.6 (77)	39.1 (43)
I did not use dietary supplements	0.0 (0)	0.0 (0)	0.0 (0)

Differences between the various pandemic waves: * *p* < 0.05, ** *p* < 0.01, *** *p* < 0.001, # multiple choice question.

## Data Availability

Excel spreadsheets with data are available from the authors.

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
