# Peer review of "Consumption of Food Supplements during the Three COVID-19 Waves in Poland—Focus on Zinc and Vitamin D"

_nutrients, 2021, doi:10.3390/nu13103361_

Round 1

Reviewer 1 Report

This paper described both the importance of food supplements during the COVID-19 waves in Poland and its related matters including education, occupation, and exercise. This study was conducted among 935 Polish and well designed, in which the meaningful questionnaire (including questions and answers) was included and the obtained data was statistically analyzed. Authors concluded several curious points form the results.

However, there is a few minor points that I wonder, and I would like to make authors add or improve the following points.

  1. In the last sentence of Abstract, what "chemical forms with good bioavailable" means more specifically? If authors aim this term to increase or enhance the bioavailability of food supplements from the gastrointestinal tract, and if its higher bioavailable rates are closely related the chemical forms and chemical species of food supplements, authors would describe just that. The same point is found in the paragraph of Conclusions.
  2. In the returned and obtained questionnaire, why 80% of the respondents were women? This results indicated that the summarized data could bias the direction of women's interest. What do authors think these important matters?

Reviewer 2 Report

Dear authors, this study could be interesting if presented with more rigor. It is difficult to embrace the entire interest of your work because of confusions. For example prevention and treatment are  physiological activities that are not allowed to food supplements. So some of your observations must be reconsidered. FS could not be used to prevent or treat a disease.

The ref 8 is dealing with molecular docking and its results are presented as physiological results, that is not fair.

The discussion must be revised because you present comparison between your results on FS consumption and experimental results using some nutrients (ex Zn but with no indication on the dosage) added to medications. It is very difficult with this kind of study to  assess what is exactly the impact of the added nutrient. Do the active principle interfere with the nutrient ? 

In my opinion using this kind of bibliography is not adequate. You cannot compare medications for treatments with FS that are not allowed to treat.

This part must be reconsidered with comparison with other elements than medication and pharmacological treatments.

The conclusion and the  end of the abstract must be changed. You introduced the selection of the appropriate forms, bioavailability... with no elements from your study, it must be improved.

Round 2

Reviewer 2 Report

dear authors, thanks to have taken into account my opinions. The second redaction is much more coherent, and much more dedicated, in the results and the discussion to FS. Then the limitation is much more considered and that improves the interest of your paper.